# Climate change and Land Use/Land Cover Change (LUCC) leading to spatial shifts in net primary productivity in Anhui Province, China

**Huan Tang**[1,2], **Jiawei Fang**[1,2], **Jing Yuan**[1,2,3]*

**1** Department of Civil Engineering, Tongling University, Tongling, China, **2** Spatial Information Acquisition and Application Joint Laboratory of Anhui Province, Tongling, China, **3** Department of Civil Engineering, Manitoba University, Winnipeg, Canada

* 154427@tlu.edu.cn

**Data Availability Statement:** The minimum dataset required to replicate our research results (mainly includes MOD13Q1 NDVI, Meteorological data, Vegetation type, LULC data and Vector

## Abstract

As an important part of terrestrial carbon cycle research, net primary productivity is an important parameter to evaluate the quality of terrestrial ecosystem and plays an important role in the analysis of global climate change and carbon balance. Anhui Province is in the Yangtze River Delta region in eastern China. Based on the theoretical basis of CASA model, this paper uses MODIS NDVI, vegetation type data, meteorological data, and LUCC to estimate the NPP of Anhui Province during 2001–2020 and analyzes its spatial-temporal pattern. The results showed that the average NPP in Anhui province was 508.95 gC· (m² ·a) $^{-1}$, and the spatial heterogeneity of NPP was strong, and the high value areas were mainly distributed in the Jiangnan Mountains and Dabie Mountains. NPP increased in most areas of Anhui Province, but decreased significantly in 17.60% of the area, mainly in the central area affected by urban and rural expansion and the transformation of the Yangtze River. The dynamic change of NPP in Anhui province is the result of climate change and land use change. Meteorological data are positively correlated with NPP. Among them, the correlation between temperature and solar radiation is higher, and the correlation between NPP and precipitation is the lowest among the three. The NPP of all land cover types was more affected by temperature than precipitation, especially forest land and grassland. The decrease of cultivated land and the increase of Artificial Surfaces (AS) may have contributed to the decrease of NPP in Anhui Province. Human activities have weakened the increase in NPP caused by climate change. In conclusion, this study refined the drivers of spatial heterogeneity of NPP changes in Anhui province, which is conducive to rational planning of terrestrial ecosystems and carbon balance measures.

## Introduction

Net Primary Productivity (NPP) refers to the energy remaining for plant growth, development and reproduction after the total organic matter synthesized by photosynthesis is subtracted from organic matter consumed by respiration. It is used to represent the carbon sequestration number of green plants [1, 2]. NPP is the net photosynthetic production produced under the

boundary data) is located in the manuscript. In addition, the MOD13Q1 NDVI, Meteorological data, Vegetation type, LULC data and Vector boundary data is third-party data, and the authors had no special access privileges to the data and that other researchers will be able to access the data in the same manner as the authors. They can be obtained in the following ways: 1. MOD13Q1 NDVI product is available from https://ladsweb.modaps.eosdis. nasa.gov. 2. Meteorological data is available from https://www.resdc.cn/. 3. Vegetation type is available from Vegetation map of the Peopleâ s Republic of China (1:1 million) ,Xinshi Zhang., Vegetation Map Editing Committee of the Chinese Academy of Sciences (https://www.plantplus.cn/ doi/10.12282/plantdata.0155), doi:10.12282/ plantdata. 4. LULC product is available from https:// www.webmap.cn/mapDataAction.do?method= globalLandCover. 5. Vector boundary data is available from https://www.webmap.cn/.

**Funding:** This study was partially funded by National Natural Science Foundation of China (42271301) and Anhui University Excellent Research and Innovation Project (No. 2022AH010094). The funders had and will not have a role in study design, data collection and analysis, decision to publish, or preparation of the manuscript.

**Competing interests:** The authors have declared that no competing interests exist.

comprehensive action of the surrounding environment and the specific physiological characteristics of plants [3–5]. It is the main factor to determine the carbon source and sink of the ecosystem and regulate the ecological process and is one of the important contents in the study of the carbon cycle process of the terrestrial ecosystem [6–11]. The intensification and spatial expansion of human activities in recent centuries has profoundly altered the world's natural and cultural landscape and has had a significant impact on ecosystem processes and their social functioning [12]. A comprehensive indicator of this global change is the mechanism of change in vegetation productivity.

As a key feature of ecosystem conditions, vegetation productivity reflects the spatial distribution and change of vegetation cover. The main climate driving factors of vegetation productivity are temperature, water supply and solar radiation [13]. NPP estimation models mainly include climate productivity model, ecological process model and light energy utilization model. Among them, the climate productivity model only considers climate factors and ignores many other factors affecting the accumulation of plant dry matter and the feedback effect of plants on the environment, so the error is large [14]. The ecological process model involves the acquisition of many complex parameters, which is very difficult. In contrast, the data acquisition of light energy utilization model is relatively difficult and has high accuracy. In addition, it can be assisted by remote sensing, saving a lot of complicated field test measurement steps. Driven by the rapid development of remote sensing technology in recent years, it has gradually become the mainstream of NPP measurement and research methods [14]. Among them, Carnegie Ames Stanford Approach (CASA) model uses remote sensing data and meteorological data to estimate NPP, which not only avoids the collection of complex parameters, but also can simulate a wide range of NPP more accurately [15]. After calibration by more than 1900 measuring stations around the world, CASA model has been widely used in the inversion simulation of carbon sequestration of different surface vegetation types in different regions at home and abroad. MOD17 is an NPP product of MODIS (Moderate-resolution Imaging Spectroradiometer), but the artificial surfaces and water in this series of data are all null values. This article needs to use statistics and analysis to examine the NPP changes of various land use types, the impact of meteorological factors on NPP of different land use types, and the impact of land use changes on NPP. However, using the CASA model, NPP for all land use types can be simulated. Therefore, this article uses the CASA model to estimate the NPP of Anhui Province.

In recent years, many scholars have used the CASA model to calculate NPP at the national scale [16–18], provincial scale [19–22], and municipal scale [23, 24]. Some scholars have also explored the relationship between NPP and climate change [25–27] or land use change [28–30]. Goroshi et al. [31] used CASA model to estimate the global NPP from 1982 to 2006 monthly, and the consistency analysis with flux tower NPP showed that Willmott's index ranges from 0.81 (Evergreen needle leaf forest) to 0.99 (Open Shrublands). Sun et al. [32] used three models to estimate NPP, and compared with the measured data, found that CASA model was more suitable for estimating NPP across China. WANG et al. [33] simulated NPP in the Weihe River Basin of China based on CASA model and discussed its spatial-temporal coverage and dynamic changes, and the results showed that NPP was related to climate parameters and altitude. Zhang et al. [34] Combining remote sensing and climate data, NPP in Beijing-Tianjin- Hebei region was simulated by CASA Model. Mahesh et al. [35] used CASA Biosphere Model in Hyderabad and Roorkee Region in India; the results also demonstrate the practicability of CASA model in estimating NPP. Grazieli et al. [36] used CASA model to calculate NPP in agricultural ecosystems in Brazil, the results showed that NPP data obtained using the CASA model and remote sensing data were in accordance with the observed data. Wu et al. [37] made use of improved CASA model simulating net primary productivity of Qinghai Lake basin alpine grassland, compared with the measured data, CASA simulates NPP with high accuracy. XU et al. [38]

estimated the NPP of urban green space by using the optimized CASA model. Chen [17] also used the CASA model to estimate the monthly NPP of Wuhan from 2017 to 2020 and evaluated the impact of meteorology on it. Yan et al. [39] took Landsat as the data source of CASA model and discussed the relationship between NPP, climate change and urbanization in Beijing. The CASA model has been validated in Europe [15, 40], Korea [41], and India [35].

Early research on the CASA model assumed maximum light energy utilization efficiency ($\varepsilon_{max}$) is an invariant constant, and later research has shown that $\varepsilon_{max}$ is related to vegetation type [42], and it is necessary to estimate the impact of water stress on plants [43]. This article uses CASA model improved by Zhu et al. [14], which incorporates vegetation type maps. $\varepsilon_{max}$ is simulated using measured NPP data from China, and calculates water stress factors using meteorological data. Previous studies have focused more on the relationship between NPP and temperature precipitation, with limited analysis of the correlation between solar radiation; However, there is even less analysis of the correlation between NPP of different land use types and three meteorological factors. The unit area NPP varies among different land use types, so the transfer between different land use types can lead to changes in NPP. The increase in low-density carbon land use types and the decrease in high-density carbon land use types will lead to a decrease in NPP. On the contrary, NPP will increase. Therefore, this paper takes Anhui Province as the research object, using the improved CASA model and ArcGIS, ENVI and other software for research and application. Using 250m resolution MODIS NDVI (normalized difference vegetation index) data, vegetation type data, meteorological data (precipitation data, temperature data, total solar radiation) and land use data from 2001 to 2020, The analysis of driving factors of spatial heterogeneity of NPP changes in Anhui province was detailed, to judge the spatial-temporal pattern of NPP and the driving force of NPP influencing factors in Anhui province.

## Materials and methods

### Study area

Anhui Province is located in East China, the Yangtze River Delta region, the terrain from north to south by plain, hills, mountains, within the Yangtze River, Huaihe River and Chaohu Lake, can be divided into three areas north of the Huai River, Jianghuai area and south of the Yangtze River (See Fig 1, Digital Elevation Model is downloaded for free from Geospatial Data Cloud, https://www.gscloud.cn/sources/accessdata/310?pid=302). Anhui Province is a transitional region between warm temperate zone and subtropical zone. North of Huaihe River is a warm temperate semi-humid monsoon climate, south of Huaihe River is a sub-hot humid monsoon climate. The average annual temperature is 14–17 ˚C, and the average annual precipitation is 773–1670 mm.

### Data

There are five main types of data.

1. Vector boundary data. One type is the vector map of the entire region of China, and the other is the administrative boundary map of Anhui province. These data are all sourced from national catalogue service for geographic information (https://www.webmap.cn/commres.do?method=result100W).

2. NDVI time series data (2001–2020). It is a grid file, derived from MODIS data, using the NASA website MOD13Q1 series products (https://search.earthdata.nasa.gov), with 16 days as the cycle of NDVI data of 250—meter resolution. It is obtained by batch clipping, transprojection, and maximum value synthesis. The NDVI data of 12 months of a year were combined to form the average annual NDVI distribution map.

3. Meteorological data. The climate dataset was derived from 77 climatological stations (Fig 1 (b)) in Anhui province, including three types of data from 2001 to 2020: temperature, precipitation, and solar radiation, and were provided by Resource and Environmental Science Data Platform (https://www.resdc.cn/data.aspx?DATAID=349). To ensure the quality of data, we delete the suspicious and missing records. Since many researches have demonstrated that the Kriging method is an effective interpolation method, having been widely used for regionalizing various variables at different scales [44, 45], these climatological data were interpolated through the Kriging method, to produce raster images. All data were resampled at a resolution of 250 m× 250 m and repeatedly projected for WGS_1984_Albers. Time range consistent with NDVI.

4. Vegetation type map is a raster file to determine the spatial distribution of various vegetation types. Based on 1:1 million Chinese vegetation type map compiled by the Vegetation Map Editing Committee of the Chinese Academy of Sciences in 2001 [46], all vegetation types were re-screened, merged, and coded. The main vegetation types in Anhui province are needle leaved forest, broadleaved forest, bush, grassland, swamp, farmland, and water body.

5. land-use and land-cover (LULC). The remote sensing data with a resolution of 30m×30m was provided by national catalogue service for geographic information (https://www.webmap.cn/mapDataAction.do?method=globalLandCover). We resampled it to a resolution of 250 meters using the nearest neighbor method based on ArcGIS 10.8 [45]. Land use types include divided into Cultivated Land, Forest, Grassland, Wetland, Water, Artificial Surfaces and Bare land.

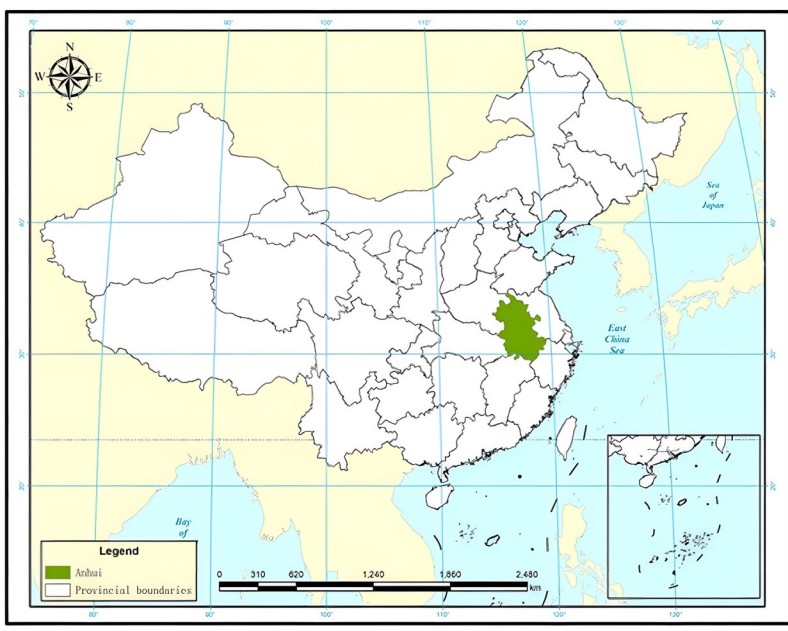

(a) location of Anhui in China.

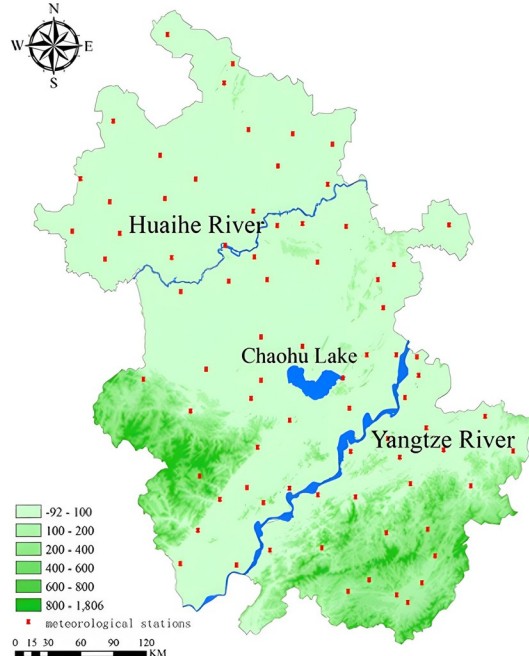

(b) Distribution map of meteorological stations in Anhui Province.

**Fig 1. Geographical location of the study area.**

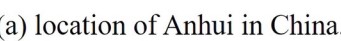

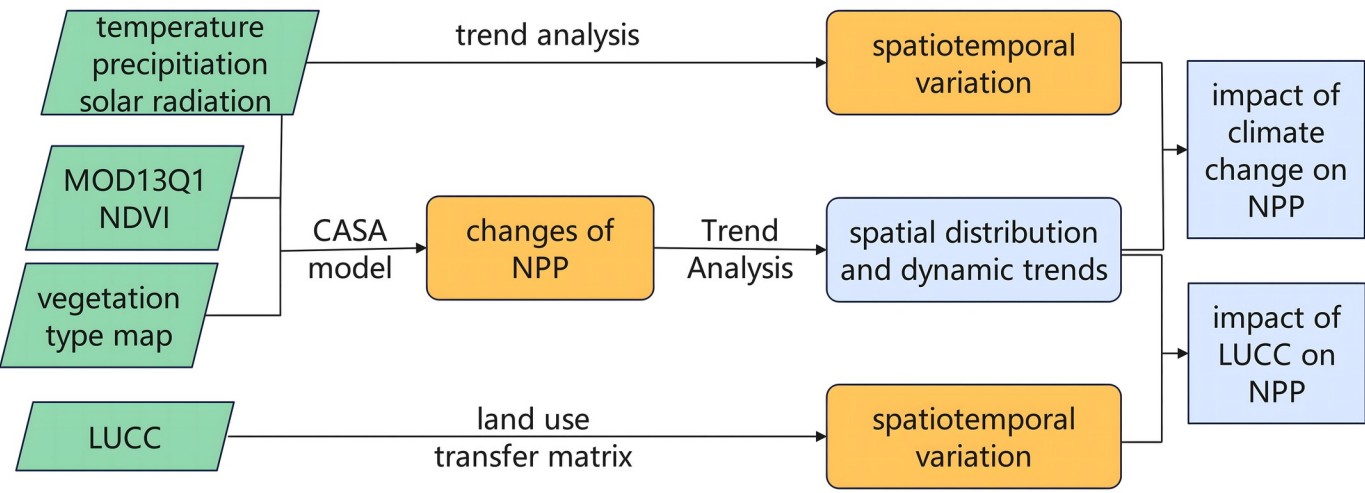

**Fig 2. The flow chart of this study.**

## Methods

As shown in Fig 2 below, this study is mainly divided into three steps. First, NPP is quantified based on long-term NDVI, vegetation type map, meteorological data, and CASA model, and compared with relevant research data and MOD17Q3 products for verification. Finally, the spatial and temporal distribution of NPP is studied by trend analysis. This step mainly analyzes the relationship between climate factors and NPP, including temperature, precipitation, and solar radiation. At the same time, the temporal and spatial changes of land use and their effects on NPP were further analyzed.

## CASA model

This study builds upon Zhu Wenquan's enhanced CASA model [14, 47], $NDVI_{max}$, $NDVI_{min}$ and $SR_{max}$ of 9 vegetation types were configured. (the maximum value of simple ratio index, Are the ratio of near infrared to infrared), $SR_{min}$, and $E_{max}$ (gC/MJ) (ideally, maximum light utilization) parameters. The calculation of $NDVI_{max}$ and $SR_{max}$ requires the vegetation type map and the maximum data of NDVI time series in Anhui province.

Run the CASA model in ENVI and import the configured parameters to generate annual and monthly net primary productivity of vegetation. Then the trend analysis was carried out in ENVI and the correlation analysis was carried out in ArcGIS.

The methodological equation of the CASA model is outlined as follows:

$$NPP(x, t) = APAR(x, t) \times \varepsilon(x, t) \tag{1}$$

The NPP measured by the model can be expressed in two key factors: the Absorbed Photosynthetic Active Radiation (APAR) and the actual light energy utilization ratio ($\varepsilon$) of the plant. $NPP(x,t)$ represents the unit area NPP in the pixel x in month t. $APAR(x,t)$ represents the photosynthetically active radiation absorbed by the pixel x in month t and $\varepsilon(x,t)$ represents the actual light energy utilization of pixel x in month t.

## Trend analysis

The trend line method of univariate linear regression can reflect the trend characteristics of pixel value changes in different periods by analyzing the size of pixel values [48]. This article

uses this method to analyze the trend of simulated NPP changes in Anhui Province from 2001 to 2020, and the calculation formula is as follows [49, 50]:

$$slope = (n * \sum_{i=1}^{n}(i * Y_i) - \sum_{i=1}^{n} i * \sum_{i=1}^{n} Y_i)/(n * \sum_{i=1}^{n} i^2 - (\sum_{i=1}^{n} i)^2) \qquad (2)$$

Slope is the trend of NPP change, i is the annual variable, n is the number of years, and $Y_i$ is the NPP value in the i-th year. When slope is greater than 0, NPP increases, otherwise it decreases.

## Simple correlation coefficient analysis

Simple correlation coefficient is the calculation of the correlation between simulated NPP values and climate factors (temperature, precipitation, and solar radiation). The calculation formula is as follows [45, 51, 52]:

$$r_{xy} = \sum_{i=1}^{n}[(x_i - \bar{x}) * (y_i - \bar{y})]/\sqrt{\sum_{i=1}^{n}(x_i - \bar{x})^2 * \sum_{i=1}^{n}(y_i - \bar{y})^2} \qquad (3)$$

$r_{xy}$ is the simple correlation coefficient between x and y, $x_i$ is the NPP value for the i-th year or month, $y_i$ is the correlation factor value for the corresponding year or month, $\bar{x}$ and $\bar{y}$ are the average NPP value and the corresponding correlation factor average, respectively, and i is the annual variable.

## Partial correlation analysis

Coefficient of partial correlation is determined based on the simple correlation coefficient. The partial correlation coefficient can be used to analyze the correlation between multiple variables, that is, to calculate the correlation coefficient between two variables after fixing one variable, which can be used in many cases to assess a said relationship [53, 54]. This article calculates the partial correlation coefficient between NPP and temperature by fixing solar radiation and precipitation. Similarly, when calculating the partial correlation coefficient between NPP and precipitation, two variables, temperature and solar radiation, were fixed. The calculation of the partial correlation coefficient between NPP and solar radiation involves two variables: fixed temperature and precipitation [54, 55]. The calculation formula is as follows [50]:

$$r_{y1,2} = (r_{y1} - r_{y2} * r_{12})/\sqrt{\left(1 - r_{y2}^2\right) * (1 - r_{12}^2)} \qquad (4)$$

$r_{y1,2}$ represents the partial correlation coefficient between variable y and variable x1 after fixed variable x2. $r_{y1}$, $r_{y2}$, and $r_{12}$ represent the correlation coefficients between y and x1, y and x2, and x1 and x2, respectively.

The trend analysis, simple correlation coefficient and partial correlation analysis were performed in the R environment with package of 'stats' (version 3.4.4) and 'ppcor' (version 1.1), respectively.

## Validation of the CASA model

Validation of the CASA Model are listed in Tables 1 and 2. As shown in Table 3, two methods are used to verify the results of CASA model. First, the results were compared with the results of relevant studies, and it was found that the NPP results measured in this study showed significant similarities with the results obtained in relevant studies. Among the published studies on NPP, the most widely used dataset is MOD17A3 [56, 57]. This dataset has been validated in research on vegetation growth, environmental monitoring, and global change [1, 58], and can

**Table 1. Validation compared to findings of relevant studies.**

| Validation reference | Measured Area | Time (Year) | NPP (gC·(m$^2$ ·a) $^{-1}$) | NPP of this study (gC·(m$^2$ ·a) $^{-1}$) |
|---|---|---|---|---|
| CASA(*42*) | Yangtze River Delta urban agglomeration | 2000–2018 | 538.69 | 508.95 |
| Based on MOD17A3(*33*) | Yangtze River Delta urban agglomeration | 2000–2020 | 510.60 | 508.95 |

**Table 2. Validation compared to the MOD17A3 data (gC·(m$^2$ ·a) $^{-1}$).**

| Time (Year) | 2001 | 2005 | 2010 | 2015 | 2020 |
|---|---|---|---|---|---|
| NPP of MOD17A3 | 426.70 | 483.37 | 441.24 | 422.39 | 427.51 |
| NPP of this study | 479.77 | 547.61 | 496.01 | 500.02 | 477.27 |

**Table 3. Data sources.**

| Data | Spatial resolution | Sources |
|---|---|---|
| Vector boundary data of China and Anhui Province | vector | https://www.webmap.cn/commres.do?method=result100W |
| MOD13Q1 NDVI | 250m×250m | https://ladsweb.modaps.eosdis.nasa.gov/search/ |
| monthly mean temperature | - | https://www.resdc.cn/data.aspx?DATAID=349, *GST_ave* |
| monthly total solar radiation | - | https://www.resdc.cn/data.aspx?DATAID=349, *SSD* |
| monthly total precipitation | - | https://www.resdc.cn/data.aspx?DATAID=349, *PRE* |
| Vegetation type | 1km×1km | https://www.plantplus.cn/doi/10.12282/plantdata.0155 |
| land-use and land-cover | 30m×30m | https://www.webmap.cn/mapDataAction.do?method=globalLandCover |

be used to validate the results of the CASA model [59]. Secondly, compared with MOD17A3 data, which was download from National Aeronautics and Space Administration (https://ladsweb.modaps.eosdis.nasa.gov/search/), the spatial resolution of this data set is 500 meters, which is 250 meters in this paper. In Table 2, it could be noted that both methods produced highly similar NPP levels. Therefore, the NPP of Anhui Province obtained by CASA model in this study is reliable and can be further analyzed on this basis.

## Results

### NPP temporal variation characteristics

From 2001 to 2020, the average annual NPP in Anhui Province showed a fluctuating upward trend, and the increase rate was 1.2258 gC·(m$^2$ ·a) $^{-1}$ (Fig 3). The multi-year average is 508.95 gC·(m$^2$ ·a) $^{-1}$), with a maximum of 564.15 gC·(m$^2$ ·a) $^{-1}$), which occurred in 2013. The lowest value, 469.28 gC·(m$^2$ ·a) $^{-1}$, occurred in 2011 (See Fig 3(a)). For the sub-regions, the area north of Huaihe River grew the fastest (See Fig 3(b)), with a growth rate of 2.2097 gC·(m$^2$ ·a) $^{-1}$). the slowest growth is in the south of the Yangtze River (See Fig 3(d)), the growth rate is only 0.1118 gC·(m$^2$ ·a) $^{-1}$. And the average annual NPP was as follows: South of Yangtze River (548.74 gC·(m$^2$ ·a) $^{-1}$)) > North of Huaihe River (479.33 gC·(m$^2$ ·a) $^{-1}$)) > Jianghuai River (479.01 gC·(m$^2$ ·a) $^{-1}$)).

Fig 4 shows the annual NPP trends of different land use types in Anhui Province. Overall, NPP showed an increasing trend regardless of land use type. Different land use types for many years, based on the relationship between the average NPP: forest (603.17 gC·(m$^2$ ·a) $^{-1}$) > grassland (487.06 gC·(m$^2$ ·a) $^{-1}$) > cultivated land (462.71 gC·(m$^2$ ·a) $^{-1}$) > artificial surface (431.68 gC·(m$^2$ ·a) $^{-1}$) > Wetland (332.32 gC·(m$^2$ ·a) $^{-1}$) > bare land (250.57 gC·(m$^2$ ·a) $^{-1}$) > water (217.64 gC·(m$^2$ ·a) $^{-1}$). Fig 5 shows that among all land use types, wetland has the highest

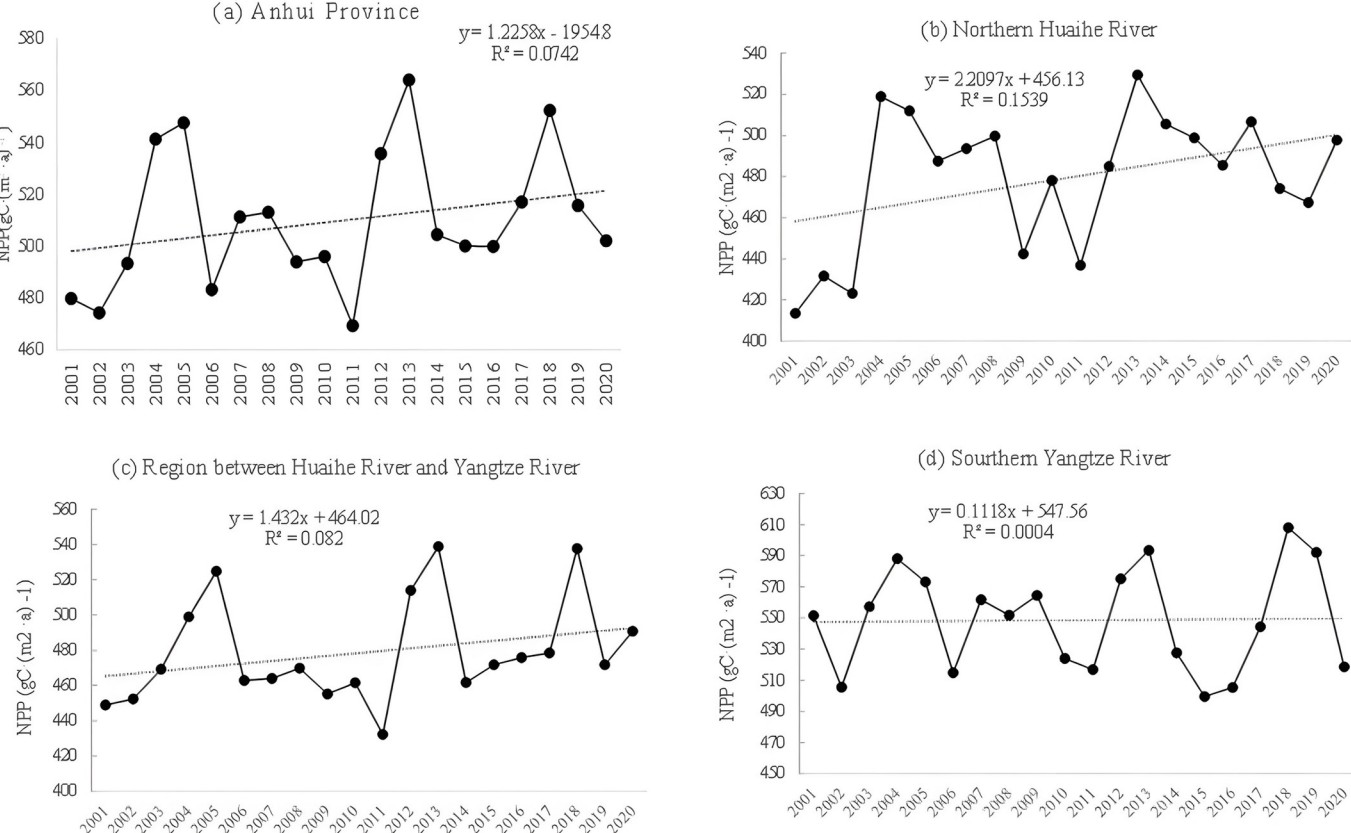

**Fig 3. Annual mean NPP and its change trend from 2001 to 2020: (a) the whole Anhui; (b) Northern Huaihe River; (c) Region between Huai River and Yangtze River; (d) Southern Yangtze River.**

annual NPP growth rate, reaching 4.6282 gC·(m$^2$·a)$^{-1}$, followed by cultivated land, water, grassland, and forest. It could be noted that in Fig 6, the average annual NPP of grassland and forest land was basically stable. The average annual NPP of artificial surface and bare land showed a downward trend. The decrease rate of NPP on artificial surface was the highest, reaching 1.6323 gC·(m$^2$·a)$^{-1}$.

The temporal variation of NPP is basically consistent with that of plant growth. Summer NPP is the highest of the year, with 13 NPP peaks occurring in July and 7 NPP peaks occurring in August. The four months from November to February of the following year are the trough range of the annual NPP. NPP has been increasing gradually since March (Fig 6).

## NPP spatial change characteristics

The spatial pattern of NPP in Anhui Province showed strong spatial heterogeneity from north to south. Fig 7 shows that the average annual NPP in the south of the Yangtze River is higher than that in the north of the Huai River and Jianghuai area. In the south of the Yangtze River, the average annual NPP range from 100 to 800 gC·(m$^2$·a)$^{-1}$, NPP below 300 gC·(m$^2$·a)$^{-1}$ is mainly distributed in the Yangtze River basin and lakes, and 300 to 500 gC·(m$^2$·a)$^{-1}$ is mainly distributed in the middle and lower reaches of the Yangtze River plain. The NPP above 500 gC·(m$^2$·a)$^{-1}$ is mainly distributed in the mountainous areas of Jiangnan, and the average annual NPP of Huangshan is above 700 gC·(m$^2$·a)$^{-1}$. The average annual NPP in the Jianghuai

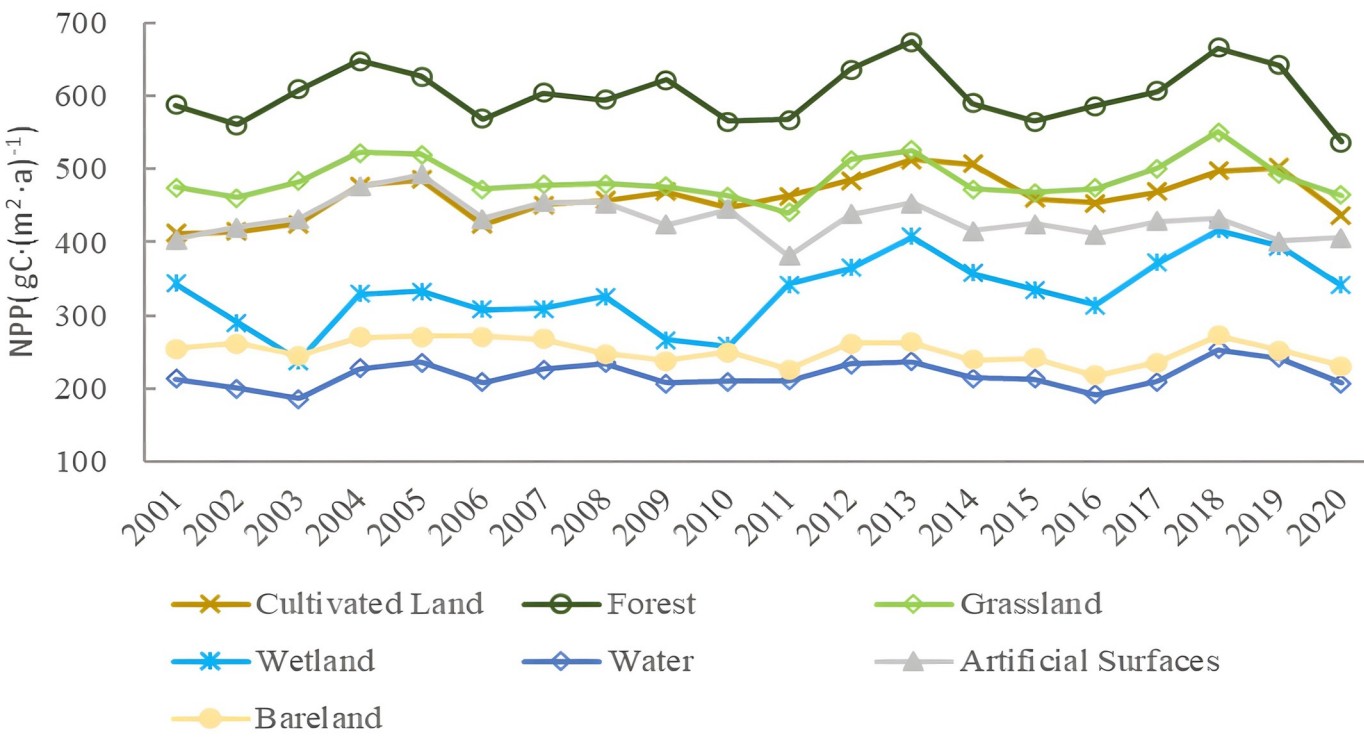

**Fig 4. Variation of NPP in different land use types.**

area fluctuates widely, and the average annual NPP in the Huaihe River basin and Chaohu Lake is less than 100 gC·(m² ·a) $^{-1}$. 100 ~300, which is mainly distributed in urban centers, such as Hefei, the capital city of Anhui Province, which is in the north of Chaohu Lake. 300–500 gC·(m² ·a) $^{-1}$ is mainly distributed in the hilly area of Jianghuai River. The NPP above 500 gC·(m² ·a) $^{-1}$ is mainly distributed in Dabie Mountains. North of Huaibei is mainly plain, except for the area where the city center is located, the average annual NPP is less than 300 gC· (m² ·a) $^{-1}$. The rest are 400~600 gC·(m² ·a) $^{-1}$.

According to the change trend and significance test results of NPP in Anhui Province from 2001 to 2020, from the perspective of spatial distribution structure (See Fig 8), NPP in most areas of Anhui Province has an upward trend, and 33.98% of the areas have a significant increase, mainly distributed in the middle of Huaibei Plain, the north of Jianghuai Hills, Dabie Mountains and the eastern part of the middle and lower reaches of the Yangtze River Plain. 27.13% of the regions increased slightly and the annual growth rate was scattered in the eastern part of Huaibei Plain, Dabie Mountains and Jiangnan Mountains. 17.60% of the areas decreased significantly, mainly distributed in the banks of the Yangtze River, reservoirs and near urban centers. 21.29% of the regions were slightly reduced, and scattered in the Yangtze River basin, the north of Huaibei Plain and the mountainous areas of Jiangnan.

## Discussion

### Spatiotemporal variation in vegetation NPP

The average NPP of the study area from 2001 to 2020 is 508.95 gC·(m² ·a) $^{-1}$, with an overall fluctuating upward trend. The average annual NPP is the highest in the south of the Yangtze River, followed by the north of the Huai River and the lowest in the Jianghuai area, which is

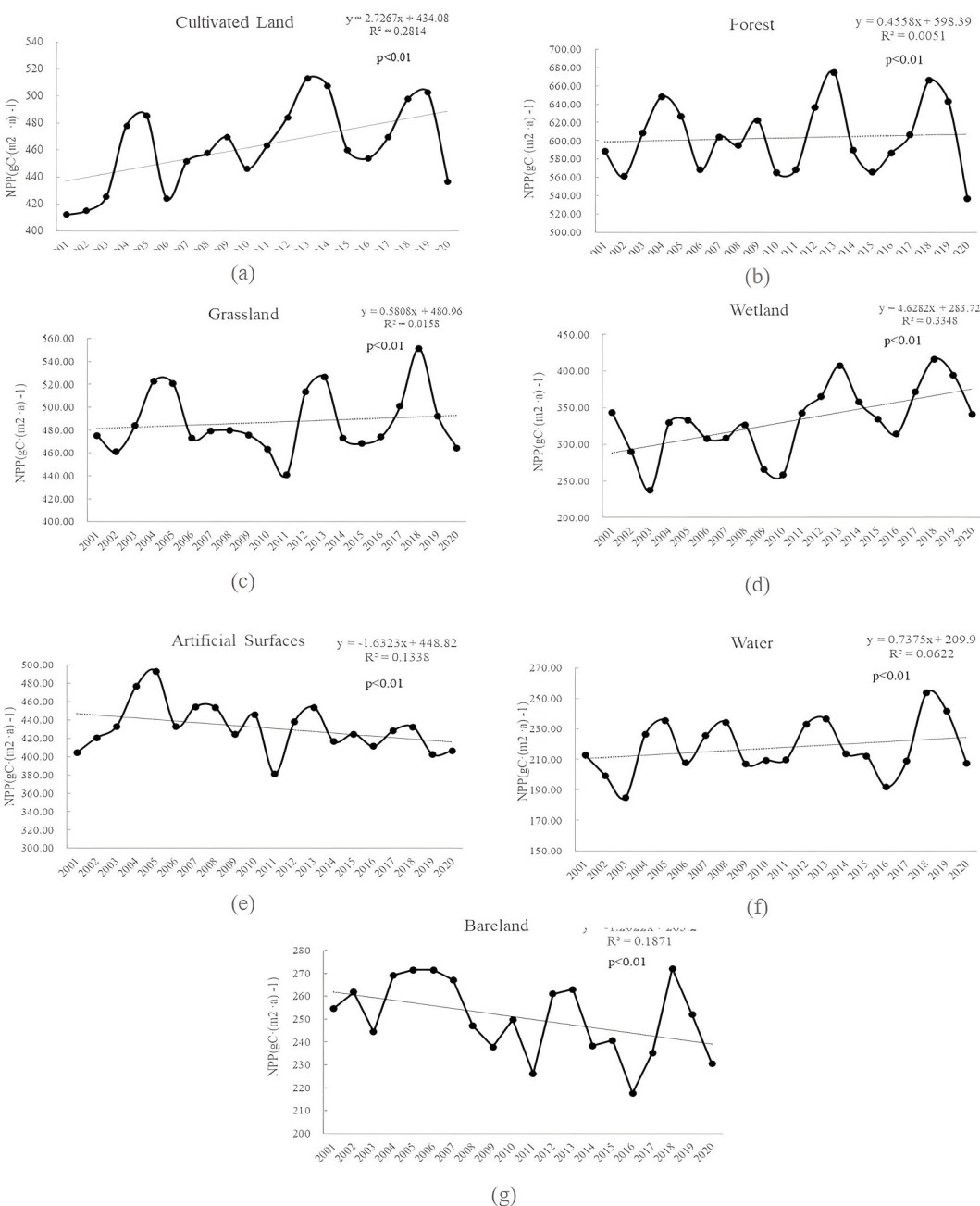

**Fig 5. Variation trend of NPP in different land use types.**

the same as the conclusion of Yang et al. [59]. LI et al. [60] showed that the NPP was generally on an increasing trend in the Yangtze River Delta region from 2000 to 2000 2019, with the average NPP value of 550.17 gC·(m$^2$·a)$^{-1}$. Nearly half of the cities in Anhui province are divided into the Yangtze River Delta, and the NPP average is smaller than LI's because the areas north of the Huai River are not part of the Yangtze River Delta, and they reduce the NPP average. The results of Wang et al. [2] and Jia et al. [61] both showed that the average annual NPP in the Yangtze River Basin showed an obvious upward trend over time. However, it also

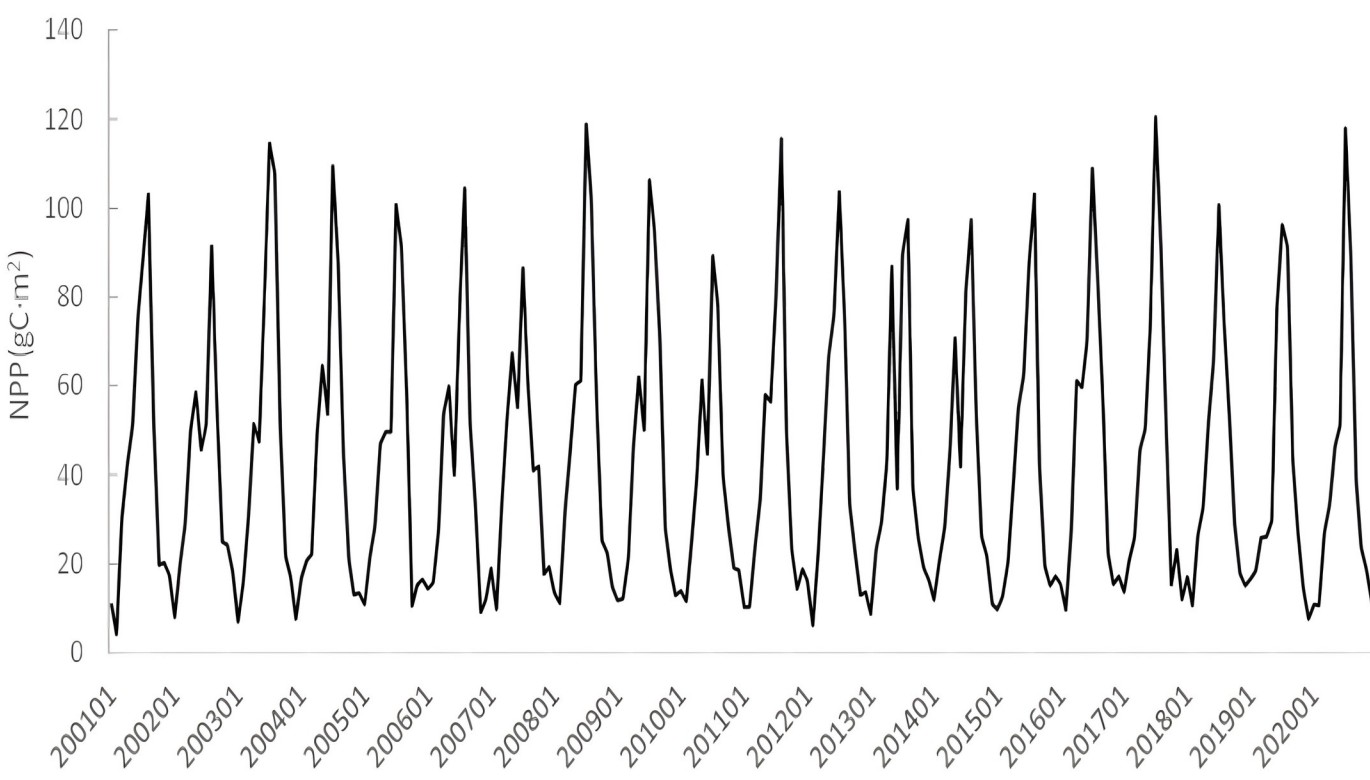

**Fig 6. Monthly NPP value changes from 2001 to 2020.**

shows that due to the influence of the Yangtze River Economic Belt, the urbanization process has accelerated rapidly, and in the cities and their surrounding areas, NPP shows a downward trend, which is consistent with the results of this paper.

Forests in Anhui province are mainly concentrated in the Dabie Mountains and the Jiangnan Mountains. The average NPP is 603.17 gC·(m² ·a) $^{-1}$. Research by Sun et al. [32] and Wang et al. [2] shows that In southern China, the evergreen broad-leaved forest is widely distributed and rich in resources, the annual average NPP is higher than 600 gC·(m² ·a) $^{-1}$. The values between 400 and 700 gC·(m² ·a) $^{-1}$ were located in the middle and lower reaches of the Yangtze River. Rich precipitation and groundwater are more conducive to the growth of vegetation. It is consistent with the conclusion of this paper. The NPP of forest land in this article has shown an upward trend over the past two decades. The upward trend of forest NPP is consistent with the increase of land vegetation NPP in China and the Northern Hemisphere [11, 62].

## Correlation analysis of annual scale NPP and meteorological factors

Generally, climate factors such as temperature, rainfall and solar radiation value directly or indirectly affect the accumulation of net primary productivity of vegetation to a certain extent. Variables cooperate or counter with each other to regulate regional NPP value, and in turn, changes in NPP value are a good reflection of climate change. In order to clarify the relationship between multiple variables, it is necessary to conduct correlation research [63, 64].

The average annual temperature in Anhui Province from 2001 to 2020 is 15.69˚C ~ 16.86˚C, with an average of 16.29˚C. It grows slowly at 0.0143˚C·a$^{-1}$. From 2001 to 2020, the

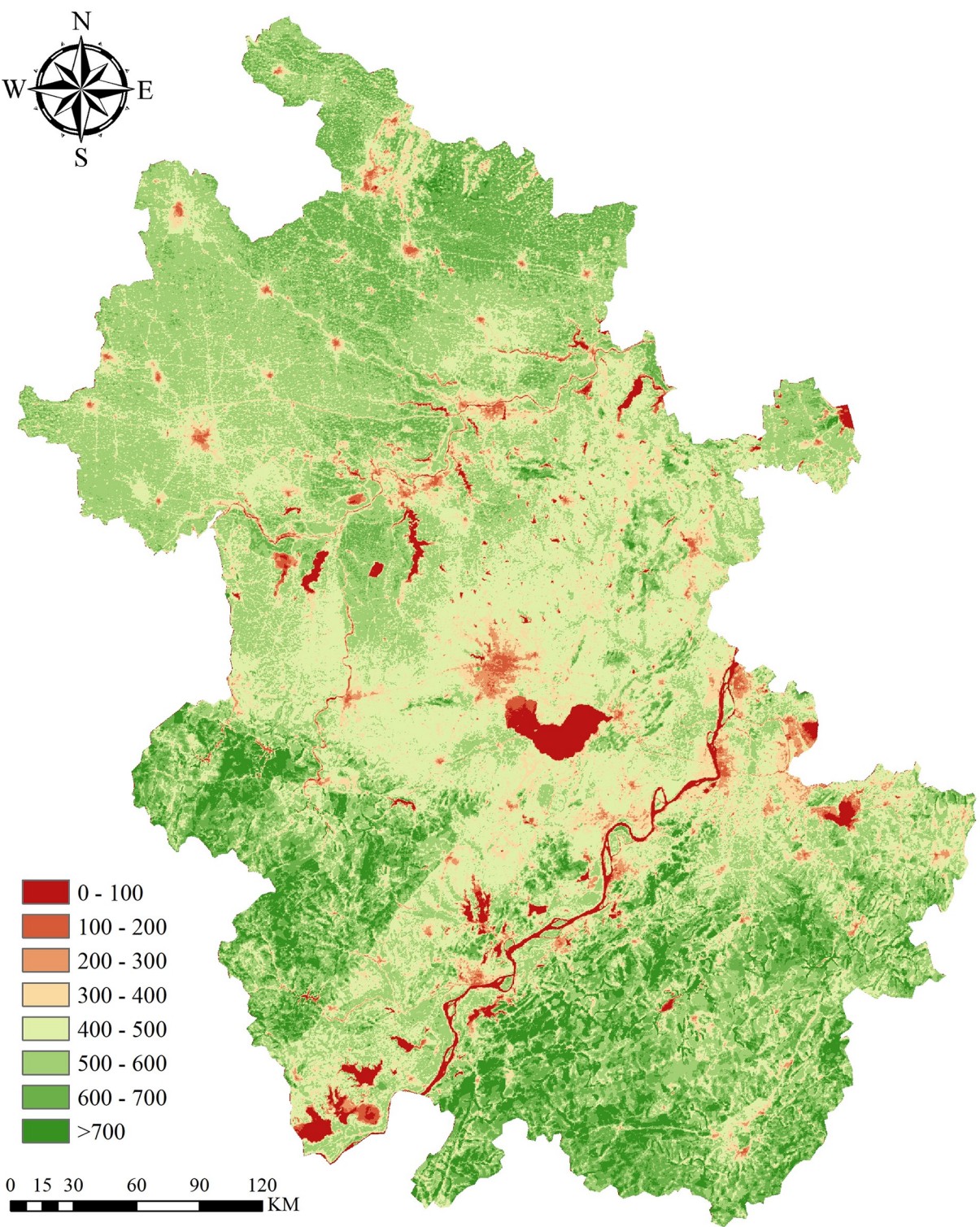

**Fig 7. Spatial pattern of the mean annual NPP calculated by CASA model.**

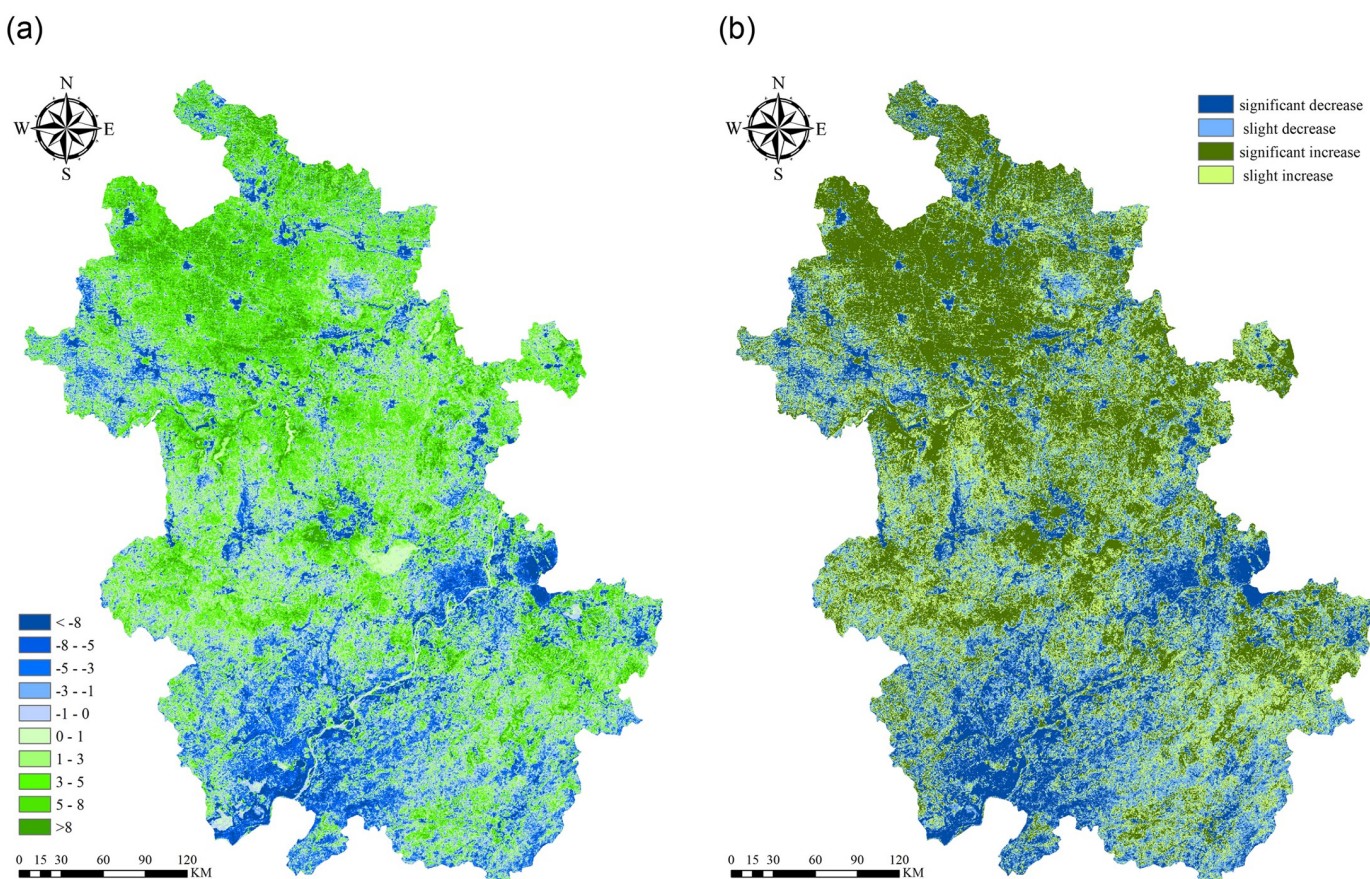

**Fig 8. Spatial distribution of NPP trend and significance of NPP trend in Anhui from 2001 to 2020.** (a) NPP trend. (b) significance of NPP.

total annual precipitation in Anhui Province ranged from 892.58 mm to 1641.89 mm, with an average of 1215.41mm. The annual growth rate was 13.166mm. Based on the temperature and precipitation data of Anhui Province, the linear correlation between inter-annual NPP and temperature and precipitation in Anhui Province was established. As can be seen from Fig 9, annual mean temperature and annual precipitation are not highly correlated with annual NPP on an annual scale. Therefore, we need to consider the impact of monthly scale data on NPP changes.

## Simple correlation coefficient analysis of monthly scale NPP and meteorological factors

The correlation between monthly scale NPP and meteorological factors is also calculated. Fig 10 shows the average correlation coefficients of monthly NPP and monthly temperature, precipitation, and solar radiation were 0.873. 0.591 and 0.669, respectively, showing that the vegetation NPP was more closely related to temperature. The proportion of regional area in the total area was 99.50%, 95.01% and 99.07%, respectively (P<0.01). Among the three factors, the significance of NPP and each factor is consistent, but the correlation difference is obvious. In addition to the Huaihe River, Yangtze River, Chaohu Lake and some reservoir areas, NPP in other areas showed a very significant correlation with temperature and solar radiation

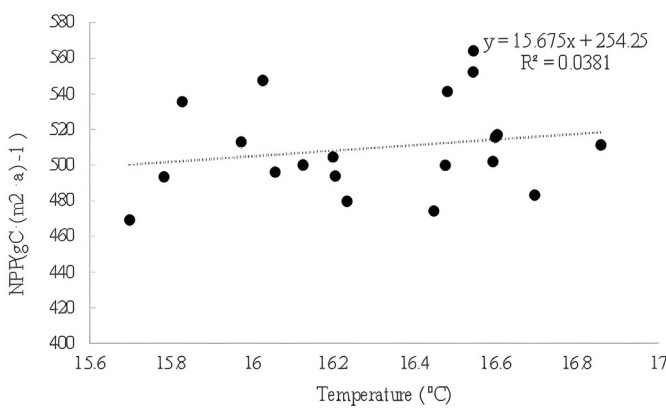

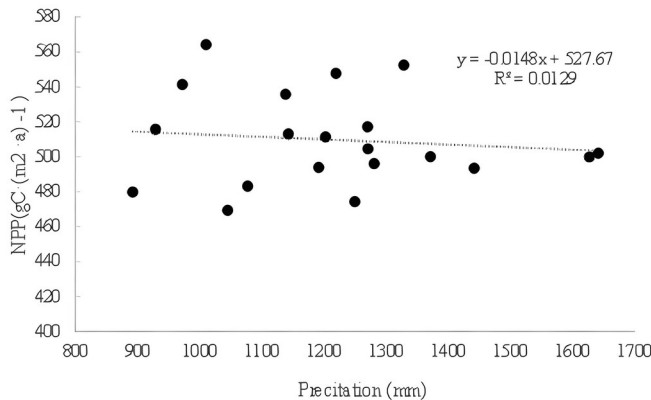

(a) The correlation between NPP and annual average temperature.

(b) The correlation between NPP and annual precipitation.

**Fig 9. Relationship between temperature, precipitation and NPP.**

(P<0.01). The correlation between NPP and precipitation was slightly weaker, but the correlation was much lower. In the Jiangnan Mountain area, especially in the southern part of Huangshan, the correlation coefficient was only about 0.15.

According to the correlation distribution map of NPP and temperature (See Fig 10(a)), south of the Yangtze River > Jianghuai area > north of the Huai River. Except for the Yangtze River, Huaihe River, Chaohu Lake and reservoir, the correlation coefficient between NPP and temperature in the south of the Yangtze River is above 0.7, with an average of 0.8174. The correlation coefficient in the Jianghuai area is above 0.6, with an average value of 0.7791, while the correlation coefficient in most areas north of the Huaihe River is 0.4~0.7. The mean value is 0.6281. Multiple studies have demonstrated an elevation in NPP associated with warming temperature, and such a positive correlation was observed in Anhui [34, 44].

From the correlation distribution map of NPP and precipitation (See Fig 10(b)), the Jianghuai region was > north of the Huaihe River > south of the Yangtze River. In addition to the Yangtze River, Huaihe River, Chaohu Lake and reservoir, the correlation coefficient between NPP and precipitation in Jianghuai region is 0.4~0.7, with an average value of 0.4766. In the north of the Huaihe River, the mean value was 0.2–0.7, and in the south of the Yangtze River, the mean value was 0.1–0.5, and the mean value was 0.3075. Wang et al. [44] showed that the changes of precipitation in the Yangtze River Delta region of China during 2000–2018 were positively correlated with the changes of NPP, and the results in this paper were consistent with his results. Precipitation is the main water source for vegetation growth and development, and appropriate precipitation helps to maintain soil moisture, which is crucial for the normal physiological activities of crop roots. In addition, the nutrients contained in rainwater play an important role in the physiology and metabolism of vegetation and are indispensable conditions for plant life systems [65].

From the correlation distribution diagram of NPP and solar radiation (See Fig 10(c)), south of the Yangtze River > North of the Huaihe River > Jianghuai River region. In the south of the Yangtze River, except for the Yangtze River system and reservoir, the rest areas are 0.6~0.8, with an average value of 0.6931. The Jianghuai region is bounded by Dabie Mountain and Chaohu Lake, and the correlation coefficient ranges from 0.6 to 0.8 in the south and from 0.4 to 0.6 in the north, with an average value of 0.5649. North of the Huai River, it is between 0.3

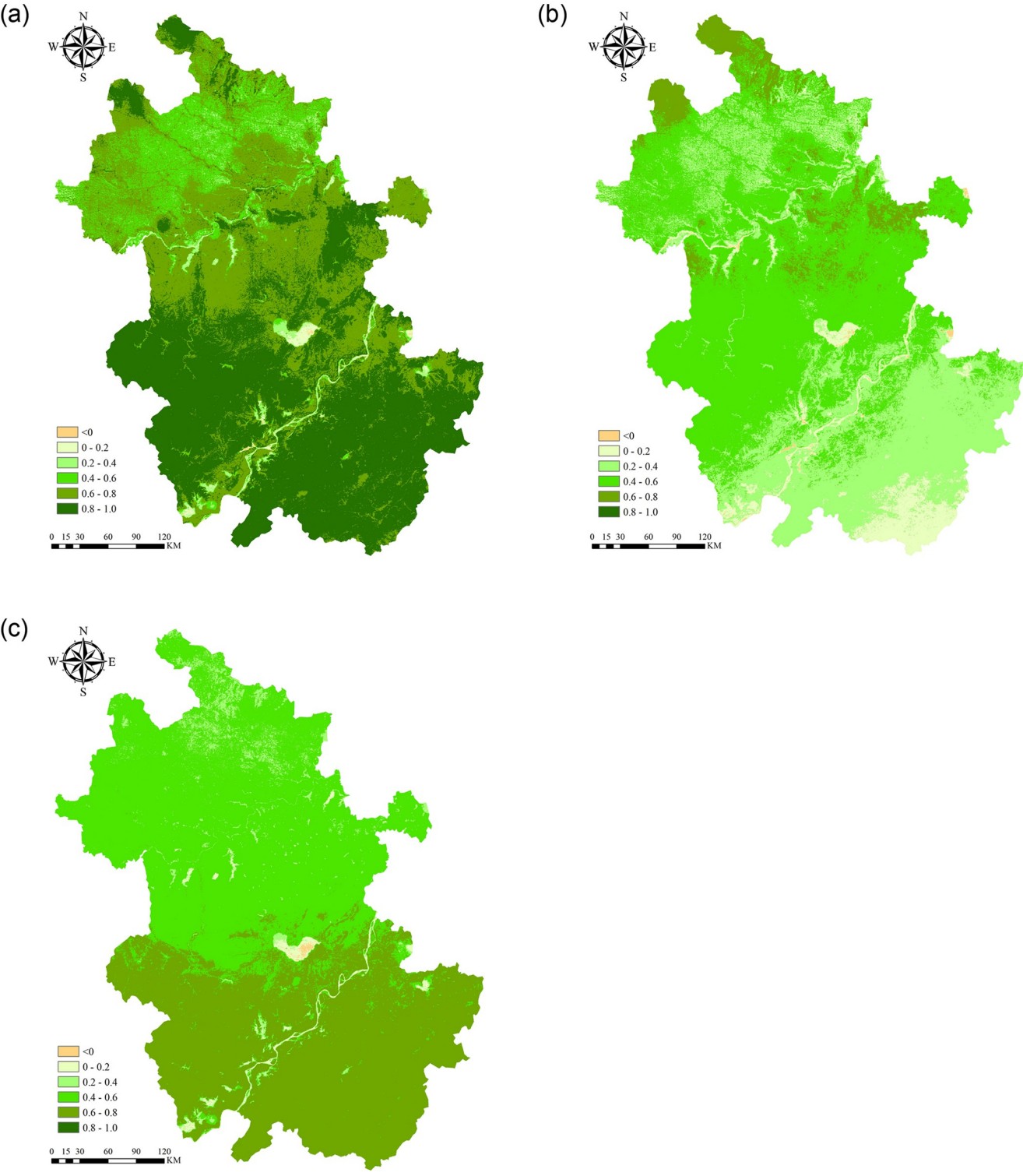

**Fig 10. Spatial distribution of the correlation coefficient between NPP and meteorological factors.** (a) correlation coefficient between NPP and temperature. (b) correlation coefficient between NPP and precipitation. (c) correlation coefficient between NPP and solar radiation.

**Table 4. Correlation coefficient between NPP and climate factors in different land cover types.**

|  | Cultivated Land | Forest | Grassland | Wetland | Water | Artificial Surfaces | Bare land |
|---|---|---|---|---|---|---|---|
| NPP-Temperature | 0.72 | 0.85 | 0.82 | 0.59 | 0.50 | 0.63 | 0.59 |
| NPP-Precipitation | 0.46 | 0.37 | 0.47 | 0.31 | 0.27 | 0.39 | 0.32 |
| NPP-Solar radiation | 0.54 | 0.69 | 0.60 | 0.47 | 0.38 | 0.43 | 0.42 |

and 0.5, with an average of 0.4566. This aligns with the findings of several studies that suggest an increase in solar radiation can boost vegetation NPP levels [61, 66]. Solar radiation is a necessary condition for vegetation photosynthesis, and suitable sunlight can promote the effective absorption, transmission, and transformation of light energy by vegetation [67]. If other conditions remain the same, the photosynthetic capacity of vegetation may also increase with increasing solar radiation [68].

Fang et al. [69] indicated that temperature is the main factor controlling changes in NPP in humid and semi-humid areas. Anhui Province belongs to the humid areas of warm temperate and subtropical regions, we also found that temperature has the highest impact on NPP, followed by solar radiation, and finally precipitation. A warm and humid climate, sufficient precipitation and heat are conducive to the increase of vegetation NPP in Anhui Province [70, 71]. Just like in most regions of the Northern Hemisphere, as temperatures rise, vegetation enters the growing season earlier and grows longer, which enhances photosynthesis and promotes vegetation coverage [72, 73].

The monthly mean NPP of different land cover types also has different feedback on monthly scale climate factors such as temperature, precipitation, and solar radiation. In Table 4, among the relationship between NPP and temperature, forest land has the greatest correlation, reaching 0.85 (P<0.1). And then the grass, Cultivated Land (0.72) > Artificial Surfaces (0.63) > Wetland \ Bare land (0.59) > Water (0.50). In the relationship between NPP and precipitation, the correlation of grassland was 0.47 (P<0.1). The correlation coefficient of cultivated land was 0.46. The correlation coefficient of other land cover types was less than 0.40. In the relationship between NPP and solar radiation, the correlation of forest land is still the largest, reaching 0.69(P<0.1). It was followed by grassland with correlation coefficient of 0.60. Then it was cultivated land > Wetland > Artificial Surfaces > Bare land > Water. And the correlation with temperature is highly consistent. It shows that temperature and solar radiation have a high degree of consistency. In general, the effect of monthly mean temperature on NPP of all land cover types was greater than that of monthly precipitation, especially for forest land and grassland. It may be because the water system in Anhui province is developed, and the carbon sequestration capacity of vegetation is more affected by temperature. Zhang et al. [74] showed in his paper that the correlation between air temperature and forest NPP in the Yangtze River Basin was stronger than that between precipitation and forest NPP, which was consistent with the results of this paper. Our results showed that precipitation had a lower correlation with forest NPP than temperature and solar radiation, this is in agreement with Du et al. [62], which indicating that precipitation has a smaller impact on forest vegetation growth than temperature and solar radiation. Our findings even showed that a positive correlation between grassland NPP and precipitation. This is in agreement with Ma et al. [50], who found that the increased precipitation could significantly increase the annual NPP of temperate grasslands.

## Partial correlation analysis of monthly scale NPP and meteorological factors

On the basis of correlation analysis, we further conduct partial correlation analysis. Fig 11(a) displayed the partial correlation distribution between NPP and temperature. The results show that after fixed precipitation and solar radiation, the average partial correlation coefficient between NPP and temperature is 0.03, range from -0.89 to 0.84. The area of the region that reached a significant level (P<0.05) only accounts for 4.34% of Anhui Province, and 44.99% of the regions showed a negative correlation. The partial correlation distribution between NPP and precipitation is presented in Fig 11(b). The average partial correlation coefficient is 0.26, and the area that reaches a significant level (P<0.05) accounts for 13.35% of the total area. After ignoring the effects of temperature and precipitation, the average partial correlation coefficient between NPP and solar radiation is 0.47, and the area ratio that reaches a significant level (P<0.05) is 37.42% (Fig 11(c)).

Previous studies have shown that warming and wetting of climate can promote vegetation growth [1, 69, 75]. However, after eliminating the effects of precipitation and solar radiation, this article found that the correlation between NPP and temperature rapidly decreased, and the area that reached a significant level also rapidly decreased. Some areas also showed a negative correlation, mainly distributed on cultivated land on both sides of the Yangtze and Huaihe Rivers. Rice is planted on both sides of the Yangtze River, and wheat is planted on both sides of the Huaihe River. High temperatures can inhibit the growth of two types of crops, which rely more on water and light hours. Du et al. [55] reached a similar conclusion when studying the partial correlation analysis between NDVI and temperature, which is that there is a negative correlation in the western and southern parts of Anhui Province. After eliminating the effects of temperature and solar radiation, the correlation between precipitation and NPP has also weakened, with negative correlations appearing on both sides of the Yangtze River and around lakes where soil moisture is abundant and excessive precipitation can inhibit vegetation growth. After eliminating the effects of temperature and precipitation, the correlation between solar radiation and NPP slightly decreased. Similar to the results of simple correlation analysis, except for water, all other regions showed a positive correlation. The explanation is that any unilateral change in climate factors is not sufficient to cause significant changes in NPP of large-scale pixel groups, but rather the result of the coupling effect of heat, water, and energy.

The response of NPP to climate factors varies among different land cover types. The partial correlation coefficients between cultivated land and temperature, precipitation, and solar radiation are 0.03, 0.29, and 0.46, respectively, with some showing a negative correlation. This may be because some crops in Anhui Province that are suitable for low-temperature growth are limited in carbon accumulation during high temperatures, while sufficient rainfall and longer sunshine hours result in more carbon accumulation. Especially in the Jianghuai region, it is greatly affected by precipitation, due to the fact that the vegetation in this area is mainly dry land. In forested areas, the partial correlation coefficients of temperature, precipitation, and solar radiation are 0.01, 0.21, and 0.55, respectively. The forest land in Anhui Province is located in relatively high-altitude mountainous areas, with a large distribution of warm and cool vegetation and strong drought resistance. The longer the sunshine hours, the more favorable it is for carbon accumulation. This is similar to Du et al. [62], who mentioned in his article that the partial correlation coefficient between forest NPP in Anhui Province and temperature and precipitation is around 0, while the partial correlation coefficient with solar radiation is around 0.45. The partial correlation coefficients between grassland and temperature, precipitation, and solar radiation are 0.03, 0.29, and 0.48, respectively. The partial correlation

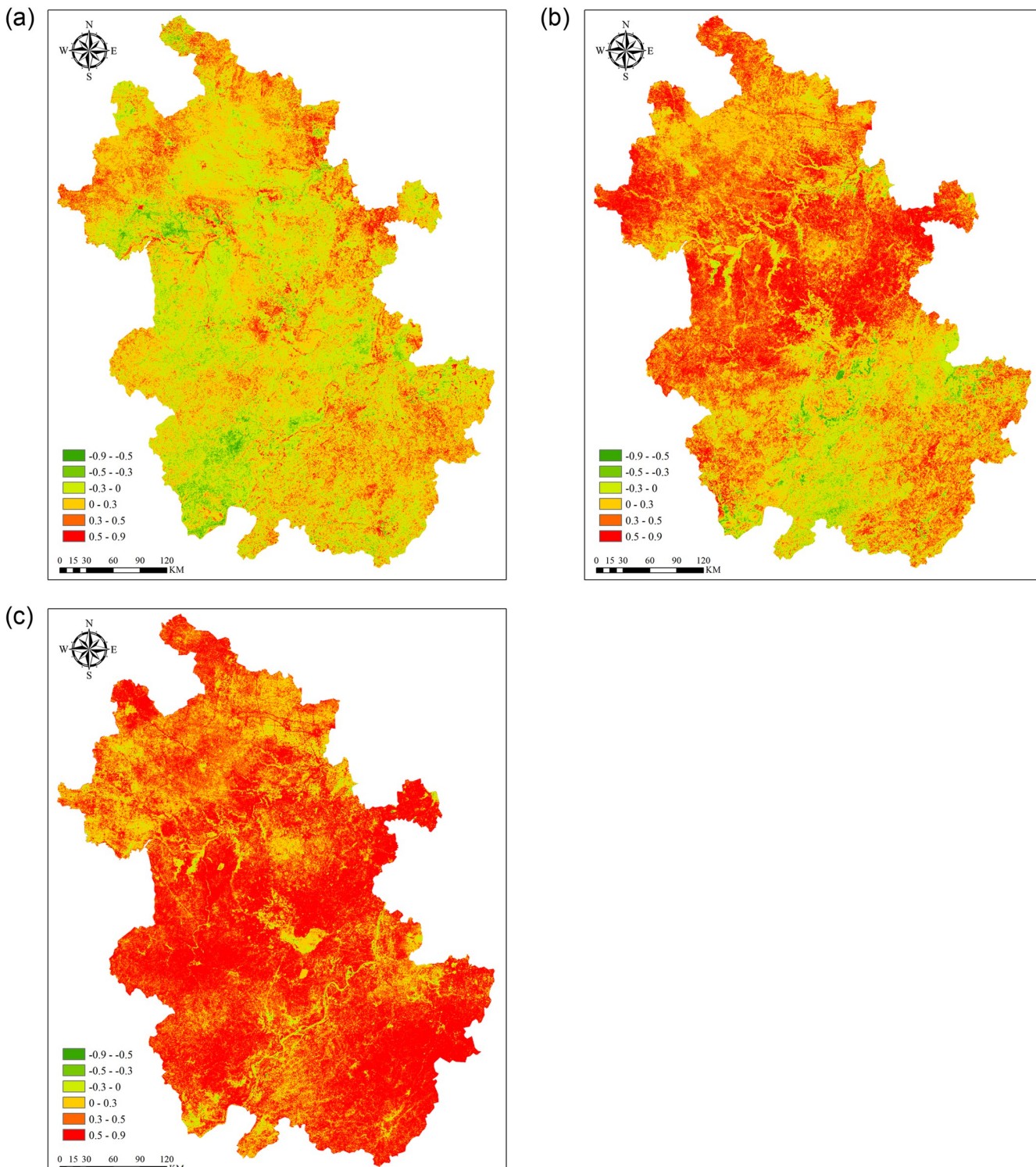

**Fig 11. Spatial distribution of the partial correlation analysis between NPP and meteorological factors.** (a) partial correlation analysis between NPP and temperature. (b) partial correlation analysis between NPP and precipitation. (c) partial correlation analysis between NPP and solar radiation.

**Table 5. Land use transfer matrix of Anhui Province from 2001 to 2020 (km²).**

| 2020 | 2001 | | | | | | | |
|---|---|---|---|---|---|---|---|---|
| | Cultivated Land | Forest | Grass Land | Wetland | Water | Artificial Surfaces | Bare land | Total |
| Cultivated Land | 72541.03 | 3359.48 | 802.33 | 110.85 | 1149.51 | 7377.80 | 9.79 | 85350.80 |
| Forest | 3392.90 | 32212.68 | 885.75 | 46.42 | 450.29 | 406.27 | 3.81 | 37398.12 |
| Grass Land | 589.82 | 821.07 | 1004.99 | 36.08 | 106.75 | 122.69 | 4.38 | 2685.78 |
| Wetland | 251.09 | 21.13 | 23.60 | 131.71 | 404.57 | 4.39 | 5.42 | 841.91 |
| Water | 1206.06 | 144.55 | 81.18 | 200.56 | 4223.79 | 113.71 | 7.44 | 5977.30 |
| Artificial Surfaces | 3317.5 | 41.15 | 44.30 | 5.95 | 60.90 | 4499.67 | 0.47 | 7969.95 |
| Bare land | 7.66 | 2.23 | 1.15 | 0 | 17.29 | 1.13 | 11.83 | 41.29 |
| Total | 81306.06 | 36602.29 | 2843.33 | 531.56 | 6413.08 | 12525.67 | 43.14 | 140265.10 |

coefficients between cities and temperature, precipitation, and solar radiation are 0.03, 0.31, and 0.42, respectively Relatively speaking, vegetation in grasslands and cities has a higher dependence on precipitation than other land cover types. Increasing precipitation can promote vegetation growth and increase the annual NPP of grasslands by increasing soil moisture in the region [76]. The more precipitation, the longer the sunshine hours, which are beneficial for carbon accumulation. Among other land cover types, the proportion of correlation with NPP reaching a significant level is relatively small. Overall, through partial correlation analysis, it was found that solar radiation has the greatest impact on the NPP of Anhui Province, followed by precipitation, and the least impact on temperature.

## NPP changes and effects of human activities

Land use and land cover change (LUCC) will directly affect the types, structures, and functions of ecosystems, and then have important impact on NPP. The most important land cover changes in Anhui Province from 2001 to 2020 was cultivated land to artificial surfaces, as well as the transformation between cultivated land and forest (See Table 5). Among all land cover types, the average annual NPP decreased most obviously, followed by bare land, and the average annual NPP increased for other land cover types. In order to understand the impact of land use change on NPP, we focused on studying the transformation from various land use types to artificial surfaces [28]. The area of artificial surfaces has increased by 4555.71 km² in the past two decades. From Table 5, it can be seen that a total of 7377.80 km² of cultivated land was converted into artificial surfaces, 406.27 km² of forest has been converted into artificial surfaces, and 122.69 km² of grassland has been converted into artificial surfaces. The unit area NPP of cultivated land, forest, and grassland are larger than that of artificial surfaces, so this transformation will cause losses in NPP. At the same time, water, wetlands, and bare land are converted into artificial surfaces, and their unit area NPP are smaller than that of artificial surfaces, which can lead to an increase in NPP. The urban expansion in Anhui Province is mainly derived from the conversion of cultivated land, so the urban expansion in Anhui Province has led to a decrease in NPP. The conclusion of Wang et al. [33] also verified the significant negative effect of urbanization on the productivity of vegetation cover system in Anhui province. Urban sprawl offsets the increase in NPP caused by climate change. The results of Yang et al. also indicate that the changes of NPP in the Yangtze River Basin are jointly influenced by LUCC changes and climate changes [59, 75].

Since the land use type in the study area is variable and the size of NPP varies with the land use type, the change between land use types will further lead to the change of NPP. Table 5 indicated the direction and area of land use changes in Anhui between 2001 and 2020. For

example, from 2001 to 2020, 3,392.90 km$^2$ and 589.82 km$^2$ of cultivated land will be converted to woodland and grassland, respectively. Since the NPP of forest land and grassland is higher than that of cultivated land, the NPP can be increased by 0.49 TgC through the above land use change. In fact, the NPP of Anhui Province showed an annual increase of 0.122 TgC, because it was also affected by urban expansion. The decrease of cultivated land and the increase of Artificial Surfaces may lead to the decrease of NPP in Anhui province, which is consistent with the results of Wang et al. [44]. Cai et al. [18] also drew a similar conclusion in his article that in northeast China, East China and central and southern China, the changes of NPP were caused by land use changes, mainly the changes of cultivated land, forest land and grassland.

The impact of human activities on vegetation ecology includes vegetation destruction and ecological construction. Since 2002, Anhui Province has implemented the project of returning farmland to forest and grassland, the project of protecting natural forest resources in key areas, the restoration of mining environment, and the construction of ecological barriers in southern and western Anhui. A series of ecological protection activities have ensured that the vegetation in Anhui Province has not been seriously damaged and stopped indiscriminate deforestation and illegal use of ecological land, resulting in the transfer of land use. The transfer of land use mode is the most direct and controllable factor of human activities on vegetation NPP.

## Uncertainty

Although this study provided a comprehensive analysis of NPP changes and climate effects in Anhui Province, there may still be some limitations. Firstly, there are deficiencies in the accuracy verification of the CASA model. Due to the inability of local relevant departments to provide on-site measurement data, this article only compares it with existing research data and MOD17A3 data. This is also one of the problems that many scholars face in studying NPP, which may cause some uncertainty [77]. This article uses NDVI and meteorological data to calculate NPP, NPP data could contain some inaccuracies which may affect the results of this study. The interpolation method of meteorological data may also affect the results of NPP [50, 52]. In addition, this article analyzes the effects of temperature, precipitation, solar radiation and urbanization on NPP, though there are some other factors (such as humidity, LUCC, etc.) that can also affect NPP. Future studies are still needed to further explore the impacts of temperature and precipitation on NPP and the effects of other climatic factors on NPP in Anhui Province.

## Conclusions

This paper takes Anhui Province as the research area, uses remote sensing data, meteorological data and LUCC, and establishes a CASA model based on the law of conservation of light energy to calculate the NPP of Anhui Province from 2001 to 2020, and analyzes the temporal and spatial distribution changes of NPP in Anhui Province in the past 20 years. The response of NPP to environmental factors in Anhui province was discussed, and the following conclusions were reached:

(1) During 2001–2020, the average value of NPP in Anhui Province was 508.95 gC·(m$^2$ ·a)$^{-1}$, and the maximum value was 564.15 gC·(m$^2$ ·a)$^{-1}$, which appeared in 2013. The lowest value was 469.28 gC·(m$^2$ ·a)$^{-1}$ in 2011. The spatial heterogeneity of NPP is strong, and the mean value of NPP is the highest in the south of the Yangtze River, and slightly higher in the north of the Huai River than between the Jianghuai River. The high value of NPP is mainly distributed in the Jiangnan Mountains and Dabie Mountains, and the average annual NPP of Huangshan is greater than 700 gC·(m$^2$ ·a)$^{-1}$.

NPP increased in most areas of Anhui Province, 33.98% of which increased significantly, mainly in the middle of Huaibei Plain, the north of Jianghuai Hills, Dabie Mountains and the

east of the middle and lower reaches of the Yangtze River plain. 27.13% of the regions increased slightly and the annual growth rate was scattered in the eastern part of Huaibei Plain, Dabie Mountains and Jiangnan Mountains. 17.60% of the areas decreased significantly, mainly distributed in the banks of the Yangtze River, reservoirs and near urban centers. 21.29% of the regions were slightly reduced, and scattered in the Yangtze River basin, the north of Huaibei Plain and the mountainous areas of Jiangnan.

(2) In simple correlation analysis, temperature, precipitation, and solar radiation are positively correlated with NPP. Among them, the correlation between temperature and solar radiation is higher, and the correlation between NPP and precipitation is the lowest among the three. In addition to Huaihe River, Yangtze River, Chaohu Lake and some reservoir areas, NPP was significantly correlated with temperature and solar radiation in other areas (P<0.01). The NPP of all land cover types was more affected by temperature than precipitation, especially forest land and grassland. In partial correlation analysis, solar radiation has the greatest impact on NPP in Anhui Province, followed by precipitation, and temperature has the smallest impact. The explanation is that any unilateral change in climate factors is not sufficient to cause significant changes in NPP of large-scale pixel groups, but rather the result of the coupling effect of heat, water, and energy.

(3) The dynamic change of NPP in Anhui Province was also affected by human activities. The most important land cover change in Anhui Province from 2001 to 2020 is the transformation of cultivated land to artificial surfaces and cultivated land and forest. The urban expansion in Anhui Province is mainly derived from the conversion of cultivated land, the NPP unit area value of cultivated land is much larger than that of urban areas, and their conversion leads to a decrease in NPP in Anhui Province. Human activities have weakened the increase in NPP caused by climate change.

In conclusion, NPP in Anhui Province showed different response mechanisms to climate, land cover and other related factors.

## Acknowledgments

Thanks to the Plant Science Data Center of Chinese Academy of Sciences (https://www.plantplus.cn) and Resource and Environmental Science Data Platform (https://www.resdc.cn/) for providing data support. Authors appreciate the reviewers for their invaluable comments which have led to significant improvement in the paper.

## Author Contributions

**Conceptualization:** Huan Tang.

**Data curation:** Jiawei Fang.

**Formal analysis:** Jiawei Fang.

**Funding acquisition:** Jing Yuan.

**Investigation:** Jiawei Fang.

**Methodology:** Huan Tang.

**Project administration:** Jing Yuan.

**Software:** Jiawei Fang.

**Supervision:** Jing Yuan.

**Validation:** Huan Tang.

**Writing – original draft:** Huan Tang.

**Writing – review & editing:** Jing Yuan.

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
