## [Decision Letter · Decision Letter 0]

12 Jun 2024

PONE-D-24-16627Climate change and Land Use/Land Cover Change (LUCC) leading to spatial shifts in Net Primary Productivity in Anhui Province, ChinaPLOS ONE

Dear Dr. Yuan,

Thank you for submitting your manuscript to PLOS ONE. After careful consideration, we feel that it has merit but does not fully meet PLOS ONE’s publication criteria as it currently stands. Therefore, we invite you to submit a revised version of the manuscript that addresses the points raised during the review process.

We look forward to receiving your revised manuscript.

Kind regards,

Dafeng Hui, Ph.D.

Academic Editor

PLOS ONE

Journal Requirements:

“The authors received no specific funding for this work.”

"The paper was partially supported by National Natural Science Foundation of China (42271301), Anhui University Excellent Research and Innovation Project (No. 2022AH010094). Authors appreciate the reviewers for their invaluable comments which have led to significant improvement in the paper."

5. Please provide a complete Data Availability Statement in the submission form, ensuring you include all necessary access information or a reason for why you are unable to make your data freely accessible. If your research concerns only data provided within your submission, please write "All data are in the manuscript and/or supporting information files" as your Data Availability Statement.

6. We note that [Figures 1, 7, 8a, 8b, 10a, 10b and 10c] in your submission contain [map/satellite] images which may be copyrighted. All PLOS content is published under the Creative Commons Attribution License (CC BY 4.0), which means that the manuscript, images, and Supporting Information files will be freely available online, and any third party is permitted to access, download, copy, distribute, and use these materials in any way, even commercially, with proper attribution. For these reasons, we cannot publish previously copyrighted maps or satellite images created using proprietary data, such as Google software (Google Maps, Street View, and Earth). For more information, see our copyright guidelines: http://journals.plos.org/plosone/s/licenses-and-copyright.

a. You may seek permission from the original copyright holder of Figures 1, 7, 8a, 8b, 10a, 10b and 10c to publish the content specifically under the CC BY 4.0 license.  

Reviewers' comments:

Reviewer's Responses to Questions

**Comments to the Author**

1. Is the manuscript technically sound, and do the data support the conclusions?

Reviewer #1: Yes

Reviewer #2: Yes

2. Has the statistical analysis been performed appropriately and rigorously? 

Reviewer #1: Yes

Reviewer #2: Yes

3. Have the authors made all data underlying the findings in their manuscript fully available?

Reviewer #1: Yes

Reviewer #2: Yes

4. Is the manuscript presented in an intelligible fashion and written in standard English?

Reviewer #1: Yes

Reviewer #2: Yes

5. Review Comments to the Author

Reviewer #1: Based on NPP data, climate data, and LUCC data, this study analyzed the spatiotemporal variation characteristics of NPP and its response to climate change in Anhui Province. The research results may contribute to further understand the relationship between NPP change and climate change. However, there are some concerns that the authors should address before it can be considered for publication.

1. Lines 32-34, I suggest the authors add more relevant references to back "It is the main factor to determine the carbon source and sink of the ecosystem and regulate the ecological process and is one of the important contents in the study of the carbon cycle process of the terrestrial ecosystem.

2. Where are the sources of data for descriptions of meteorological environmental factors in the study area?

3. How do the authors account for missing values in meteorological data?

4. I suggest the authors add a diagram about the spatial distribution of meteorological stations in Anhui Province.

5. How does the authors account for the inconsistency of spatial resolution between different data sets?

6. More mechanistic explanations should be added to further explain the relationship between NPP and climate factors.

7. In the uncertainty, I suggest the authors further discuss the uncertainty of remote sensing data including NPP data (e.g., Shen et al., 2022; Ma et al., 2022) which may affect the research results.

8. In order to further highlight the innovation of this article, it is better to compare the results of this study with some related studies.

9. Why does the authors not consider the partial correlation method to measure the correlation between meteorological factors and NPP?

References:

Asymmetric impacts of diurnal warming on vegetation carbon sequestration of marshes in the Qinghai Tibet Plateau. Global Biogeochemical Cycles, 2022, 36(7): e2022GB007396.

Spatiotemporal change of net primary productivity and its response to climate change in temperate grasslands of China. Frontiers in Plant Science, 2022, 13: 899800.

Reviewer #2: Overall, this paper has a clear research framework and is a well-structured academic paper. However, I have a few main questions:

My biggest question is, since the author uses MODIS NPP data to validate the results of the CASA model, why not directly use the MODIS product?

Furthermore, there are many similar studies, even on national and global scales. What is the significance or innovation of this study? This needs to be highlighted in the introduction and fully referenced in the discussion section. For example:

Du, Z.; Liu, X.; Wu, Z.; Zhang, H.; Zhao, J. Responses of Forest Net Primary Productivity to Climatic Factors in China during 1982–2015. Plants 2022, 11, 2932. https://doi.org/10.3390/plants11212932;

Shang, E.; Xu, E.; Zhang, H.; Liu, F. Analysis of Spatiotemporal Dynamics of the Chinese Vegetation Net Primary Productivity from the 1960s to the 2000s. Remote Sens. 2018, 10, 860;

Li, J.; Wang, Z.; Lai, C.; Wu, X.; Zeng, Z.; Chen, X.; Lian, Y. Response of net primary production to land use and land cover change in mainland China since the late 1980s. Sci. Total Environ. 2018, 639, 237–247.

Is the land use data dynamic? How did the author consider Land Cover Change as an independent variable?

The research methods section should provide specific methods for "Climate change and Land Use/Land Cover Change leading to spatial shifts in Net Primary Productivity."

Figures 1, 5, and 9 in the text should be numbered, and the p-value should be provided in Figure 5.

6. PLOS authors have the option to publish the peer review history of their article (what does this mean?). If published, this will include your full peer review and any attached files.

Reviewer #1: No

Reviewer #2: No

---

## [Author Response · Author response to Decision Letter 0]

3 Jul 2024

Dear Editors,

 I hope you are doing very well. The comments of the reviewer are very much appreciated and helped improve the manuscript significantly. We responded to your comments and made all the requested changes in a eseparate file labeled 'Response to Reviewers'. Those changes are highlighted with red color in the revised manuscript with changes marked. In the following section, we explained in detail how we responded to each of the comments. Thanks again for your valuable comments. Stay healthy and best wishes. We really appreciated your help. If you have any other concern, please don’t hesitate to let us know. I look forward to hearing from you very soon. 

Warmest Regards,

Jing Yuan

---

## [Editor Report · Decision Letter 1]

8 Jul 2024

Climate change and Land Use/Land Cover Change (LUCC) leading to spatial shifts in Net Primary Productivity in Anhui Province, China

PONE-D-24-16627R1

Dear Dr. Yuan,

We’re pleased to inform you that your manuscript has been judged scientifically suitable for publication and will be formally accepted for publication once it meets all outstanding technical requirements.

Kind regards,

Dafeng Hui, Ph.D.

Academic Editor

PLOS ONE

Additional Editor Comments (optional):

The authors have made great efforts and addressed the reviewers' concerns.
---

## [Editor Report · Acceptance letter]

10 Jul 2024

PONE-D-24-16627R1 

PLOS ONE

Dear Dr. Yuan, 

I'm pleased to inform you that your manuscript has been deemed suitable for publication in PLOS ONE. Congratulations! Your manuscript is now being handed over to our production team.

Kind regards, 

on behalf of

Dr. Dafeng Hui 

Academic Editor

PLOS ONE